# Alterations of Glycosphingolipid Glycans and Chondrogenic Markers during Differentiation of Human Induced Pluripotent Stem Cells into Chondrocytes

**DOI:** 10.3390/biom10121622

**Published:** 2020-12-01

**Authors:** Liang Xu, Hisatoshi Hanamatsu, Kentaro Homan, Tomohiro Onodera, Takuji Miyazaki, Jun-ichi Furukawa, Kazutoshi Hontani, Yuan Tian, Rikiya Baba, Norimasa Iwasaki

**Affiliations:** 1Department of Orthopaedic Surgery, Faculty of Medicine and Graduate School of Medicine, Hokkaido University, Kita 15, Nishi 7, Kita-ku, Sapporo, Hokkaido 060-8638, Japan; xuliang811026@gmail.com (L.X.); k.houman@med.hokudai.ac.jp (K.H.); takuzimiyazaki@gmail.com (T.M.); up.all.night20@gmail.com (K.H.); tyfree1001@gmail.com (Y.T.); baba76767688@gmail.com (R.B.); niwasaki@med.hokudai.ac.jp (N.I.); 2Department of Advanced Clinical Glycobiology, Faculty of Medicine and Graduate School of Medicine, Hokkaido University, Kita 21, Nishi 11, Kita-ku, Sapporo, Hokkaido 001-0021, Japan; h_hanamatsu@med.hokudai.ac.jp; 3Global Station for Soft Matter, Global Institution for Collaborative Research and Education (GSS, GI-CoRE), Hokkaido University, Kita 21, Nishi 11, Kita-ku, Sapporo, Hokkaido 001-0021, Japan

**Keywords:** cartilage injury, glycomics, glycosphingolipid, human induced pluripotent stem cells, chondrocytes, aminolysis-SALSA

## Abstract

Due to the limited intrinsic healing potential of cartilage, injury to this tissue may lead to osteoarthritis. Human induced pluripotent stem cells (iPSCs), which can be differentiated into chondrocytes, are a promising source of cells for cartilage regenerative therapy. Currently, however, the methods for evaluating chondrogenic differentiation of iPSCs are very limited; the main techniques are based on the detection of chondrogenic genes and histological analysis of the extracellular matrix. The cell surface is coated with glycocalyx, a layer of glycoconjugates including glycosphingolipids (GSLs) and glycoproteins. The glycans in glycoconjugates play important roles in biological events, and their expression and structure vary widely depending on cell types and conditions. In this study, we performed a quantitative GSL-glycan analysis of human iPSCs, iPSC-derived mesenchymal stem cell like cells (iPS-MSC like cells), iPS-MSC-derived chondrocytes (iPS-MSC-CDs), bone marrow-derived mesenchymal stem cells (BMSCs), and BMSC-derived chondrocytes (BMSC-CDs) using glycoblotting technology. We found that GSL-glycan profiles differed among cell types, and that the GSL-glycome underwent a characteristic alteration during the process of chondrogenic differentiation. Furthermore, we analyzed the GSL-glycome of normal human cartilage and found that it was quite similar to that of iPS-MSC-CDs. This is the first study to evaluate GSL-glycan structures on human iPS-derived cartilaginous particles under micromass culture conditions and those of normal human cartilage. Our results indicate that GSL-glycome analysis is useful for evaluating target cell differentiation and can thus support safe regenerative medicine.

## 1. Introduction

The regeneration of articular cartilage remains a major challenge in the field of orthopedics. This problem is important because articular cartilage has very limited intrinsic healing potential; consequently, injuries to this tissue can eventually lead to osteoarthritis [1]. Over the past decade, cell-based technologies such as autologous chondrocyte implantation (ACI) have been widely performed to repair hyaline-like cartilage [2], but several obstacles to wider adoption persist: the need to sacrifice healthy cartilage, the requirement for a two-step surgery with an initial harvest of cartilage, the difficulty of acquiring sufficient numbers of cells, and the low proliferation capacity of chondrocytes in cartilage implantation [3]. Human induced pluripotent stem cells (iPSCs), which are capable of unlimited proliferation and differentiation into any cell type in the human body, have attracted attention as a promising source of cells for regenerative medicine [4]. iPSCs can differentiate into chondrogenic lineages in vitro, and iPSC-derived cartilaginous particles can repair articular cartilage defects in vivo [5,6,7]. Hence, iPSCs are a potentially useful source of cells for cartilage regenerative therapy.

The predominant technologies for evaluation of chondrogenic differentiation of iPSCs are based on chondrogenic markers such as Sox9, type II collagen, and aggrecan, along with the histological analysis of extracellular matrix (ECM) components [5,6,7]. Recently, Gong et al. indicated that nuclear factor of activated T-cells (NFATc1) may interact with the master transcriptional regulator Sox9 and be involved in chondrogenesis regulation [8]. However, Sfougataki et al. reported that type II collagen was majorly produced in bone marrow-derived mesenchymal stem cell (BMSC) derived micromasses; in contrast, it was not detected in iPSC-derived mesenchymal stem cells derived micromasses [9]. Thus, evaluation of chondrogenic differentiation by genetic markers alone may not be sufficient. Despite the fact that histological analysis of extracellular matrix (ECM) components is also usually performed for evaluation of cartilage formation [7], it remains unclear what extent of matching is required for acceptable allogeneic transplantation of chondrocytes or cartilage. To achieve a more comprehensive and diverse evaluation of chondrogenic differentiation, new methods must be developed.

Cell-surface glycans such as glycosphingolipids (GSLs) and glycoproteins have important biological characteristics and functions, and these molecules vary widely depending on cell type and growth conditions [10]. For example, expression levels of high-mannose type N-glycans are significantly up-regulated at the late stage of differentiation of mouse chondroprogenitor cells (ATDC5), and five glycoproteins including high-mannose type N-glycans are sensitive differentiation markers of human chondrocytes [11]. During the process of chondrocyte hypertrophy, the levels of GM3, the most abundant ganglioside in chondrocytes, decrease gradually [12]. Likewise, the glycan profiles of human embryonic stem cells (ESCs) and iPSCs are also highly cell type-specific. For instance, undifferentiated human ESCs and iPSCs are coated by unique GSLs such as stage specific embryonic antigen-3 (SSEA-3), SSEA-4, Globo H, and H type 1 antigen, which serve as stem cell markers [13]. Furthermore, expression of human iPSC surface glycans changes in a characteristic manner upon differentiation. The expression level of globo- and (n) lacto-series GSLs diminishes rapidly upon differentiation of iPSCs, whereas expression of ganglioside GSLs such as GM3, GM2, and GD3 is elevated [14]. These results suggest that GSL-glycan profiles are highly cell type-specific, and that cellular GSL-glycomic analysis could be useful for evaluating the chondrogenic differentiation of iPSCs. 

In this study, we performed quantitative analysis of cell markers and the GSL-glycome of undifferentiated iPSCs, iPSC-derived mesenchymal stem cell like cells (iPS-MSC like cells), and iPS-MSC-derived chondrocytes (iPS-MSC-CDs) using the glycoblotting method combined with endoglycoceramidase digestion. Our results reveal that characteristic alterations of the GSL-glycome, as well as genetic markers and ECM components, could be evaluated during the process of chondrogenic differentiation from human iPSCs. Moreover, the glycomic approach reveals that the GSL-glycan profile of iPS-MSC-CDs is similar to that of human cartilage.

## 2. Materials and Methods

### 2.1. iPSCs: Culture And Maintenance

Human iPSC lines HPS0063 201B7 and HPS0328 606A1 (RIKEN BioResource Research Center, Tsukuba, Japan) were cultured on a feeder layer of mitotically inactivated (mitomycin C; Sigma-Aldrich, St. Louis, MO, USA) mouse embryonic fibroblasts (Oriental Yeast, Tokyo, Japan) in iPSC medium and passaged two or three times according to the subculture protocol of the Center for iPS Cell Research and Application (CiRA) [15]. iPSC medium consisted of DMEM/HAM, F12 (Gibco/Thermo Fisher Scientific, Waltham, MA, USA) supplemented with 20% knock-out serum replacement (KSR; Gibco), 1% MEM non-essential amino acids (NEAA100X; Gibco), 1% L-glutamine (0.2 M), 0.11 mM 2-mercaptoethanol (Gibco), 0.5% penicillin/streptomycin (10 U/L and 10 mg/L, respectively; FUJIFILM Wako Pure Chemical, Osaka, Japan), and 4 ng/mL recombinant human basic fibro-blast growth factor (bFGF, FUJIFILM Wako Pure Chemical). Under these conditions, iPSCs remained in an undifferentiated state, as assessed by blue alkaline phosphatase (Blue AP) staining (Vector^®^ Blue AP Substrate kit, SK-5300; Vector Labs, Burlingame, CA, USA). The iPSCs were transferred to feeder-free conditions in which vitronectin coating (Gibco) and Essential 8 medium (Gibco) were used for continuous culture, as previously described [16]; these cells were defined as iPSC feeder-free passage 0 (iPSCs (FFP0)). iPSCs (FFP0) were maintained and passaged three times as previously described [16].

### 2.2. Derivation of Mesenchymal Stem Cell Like Cells from iPSCs

Mesenchymal stem cell-like cells were induced from iPSCs as previously described [17] with minor modifications, using feeder-free iPSCs. iPSCs were seeded onto gelatin-coated 6-well plates at 5 × 10^4^ cells/cm^2^ in E8 medium with RevitaCell supplement (Gibco). At day 2, when most clumps of small cells were adhered, the medium was replaced with MSC induction medium consisting of DMEM (high glucose) medium (FUJIFILM Wako Pure Chemical), 10% fetal bovine serum (FBS; Gibco), 1% MEM non-essential amino acids (NEAA-100X; Gibco), 1% penicillin/streptomycin, and 5 ng/mL human recombinant bFGF (FUJIFILM Wako Pure Chemical). These cells were defined as passage 0 iPS-MSC like cells. When the cells reached > 90% confluence at days 6–8, they were detached with TrypLE Express (Gibco) and subsequently passaged six times at 1 × 10^4^ cells/cm^2^ on non-gelatin-coated flasks in MSC induction medium. iPS-MSC like cells at passage 6 were used for chondrogenic differentiation. MSCs (PT-2501; Lonza, Basel, Switzerland) derived from bone marrow were pre-cultured and passaged three times and used as control.

### 2.3. Chondrogenic Differentiation of BMSCs and iPS-MSC Like Cells

High-density micromass culture using bone marrow-derived mesenchymal stem cells (BMSCs; passage 3) and iPS-MSC like cells (passage 6) was performed as previously described [17]. Cells were seeded in 24-well plates at a density of 2.0 × 10^5^ cells/well in a final volume of 10 μL/drop, and then incubated for 2 h at 37 °C to allow adherence. At that time, 1 mL MSC induction medium was added to each well. These cells were defined as BMSC-derived chondrocytes (BMSC-CDs) and iPS-MSC-CDs day 0. After cultivation for 24 hours, the culture medium was replaced with chondrogenic medium. These cells were defined as BMSCs and iPS-MSC-CDs day 1. On day 2 of cultivation, the culture medium was replaced with chondrogenic medium containing human recombinant bone morphogenetic protein-2 (BMP-2) at a final concentration of 100 ng/mL (14-8507; eBioscience/Thermo Fisher Scientific, Waltham, MA, USA). Medium treated with human recombinant BMP-2 was replaced three times per week until day 21 of cultivation. Cells were harvested and analyzed at four time points (days 1, 7, 14, and 21).

### 2.4. RNA Isolation and Quantitative Real-Time Reverse Transcription-Polymerase Chain Reaction

Total RNA was extracted from each sample using Trizol reagent (Invitrogen, Carlsbad, CA, USA). For synthesis of cDNA, reverse transcription was performed using the QuantiTect Reverse Transcription Kit (Qiagen, Venlo, Nederland), and then triplicate cDNA samples were subjected to quantitative real-time polymerase chain reaction (PCR) using SYBR Green Master Mix (Finnzymes, Vantaa, Finland) on a Thermal Cycler Dice Real-Time System II (model TP900; Takara Bio, Otsu, Japan). The primers used for quantitative real-time PCR are listed in Appendix A. Gene expression levels were normalized against the corresponding levels of glyceraldehyde 3-phosphate dehydrogenase (GAPDH) mRNA.

### 2.5. Flow Cytometry Analysis

To detect mesenchymal stromal cell properties, iPS-MSC like cells were harvested and treated using the human MSC analysis kit (562245; BD Biosciences, Franklin Lakes, NJ, USA), which includes the positive markers CD44, CD73, CD90, and CD105; and the negative markers CD11b, CD19, CD34, CD45, and Human Leukocyte Antigen-DR isotype (HLA-DR) [18]. The cells were analyzed by flow cytometry on a BD FACS Canto II. Data were analyzed using the ACS Diva software (BD Biosciences) and the FlowJo data analysis package (Tree Star; Ashland, OR, USA).

### 2.6. Histological Analysis

Alcian blue is a cationic dye that forms insoluble complexes with acidic glycosaminoglycans, aiding in the quantitative determination of glycosaminoglycans; it is commonly used for the staining of cartilage [19]. Samples of iPS-MSC-CDs harvested on days 1, 7, 14, and 21 were stained with 0.1% Alcian blue 8GX solution, pH 2.5 (FUJIFILM Wako Pure Chemical).

### 2.7. GSL-Glycan Analysis

After six washes with cold phosphate-buffered saline (PBS) to remove the culture medium, samples of iPSCs, BMSCs, iPS-MSC like cells, iPSC-MSC-CDs, and (BMSC-CDs) were detached with a cell scraper and collected in a microtube (~2 × 10^6^ cells). Cells from each sample were homogenized using either a beads crusher (TAITEC, Saitama, Japan) or a BIORUPTOR II instrument (Sonic Bio, Kanagawa, Japan), and the supernatant was collected for GSL-glycomic analysis by ethanol precipitation. Intact glycans were released from GSLs by *Rhodococcus* endoglycoceramidase I (EGCase I) digestion as previously described [20]. GSL-glycans were purified by glycoblotting combined with either methylesterification [21] or aminolysis-sialic acid linkage specific alkylamidation (aminolysis- SALSA) [22].

### 2.8. Human Cartilage Preparation

Acquisition and use of patient tissues were approved by the institutional review board (IRB) of Hokkaido University (approval number: 014-0144), and informed consent was obtained in advance. Human cartilage specimens were harvested from the femoral head of patients under the age of 80 who were treated for bipolar hip arthroplasty (BHA) or total hip arthroplasty (THA) for femoral neck fracture. Exclusion criteria were osteoarthritis, rheumatoid arthritis, femur head necrosis, or infectious disease. Immediately upon receipt, each cartilage specimen was evaluated macroscopically by at least two veteran doctors to ensure that it had not suffered from damage or degeneration, and then stored at 0 °C. Cartilage specimens were treated within 9 hours of harvest in the operating room. Isolated human cartilage was homogenized with a Polytron blender (Kinematica, Luzern, Switzerland), and then cold ethanol was added to prepare the cellular lipid fraction. Intact GSL-glycans were analyzed as previously described [20].

### 2.9. Statistical Analyses

All data are presented as means ± SD of three samples from independent experiments. When two or more groups were involved, statistical comparisons were performed by ANOVA with Tukey’s post-hoc analysis using the JMP Pro software (v14.1; SAS Institute, Tokyo, Japan). Statistical significance was set at *p* < 0.05.

## 3. Results

### 3.1. Human iPSCs: Culture and Maintenance

Although maintenance of human iPSCs typically requires the absence of a murine embryonic fibroblast (MEF) feeder layer to further enhance human iPSC adhesion and growth, iPSCs cultured on a feeder layer might increase the risk of MEF contamination. Recently, a new iPSC culture approach was developed using E8 medium and a vitronectin-coated plate and was allowed to improve derivation efficiencies of feeder-free human iPSCs [16]. First, we cultured undifferentiated human iPSCs with or without a feeder layer and analyzed markers of undifferentiated cells. Typical iPSC colony formation was confirmed by microscopic observation under both conditions, as shown in Figure 1A. Blue AP staining was observed in both iPSC lines, confirming their undifferentiated state. The relative expression levels of well-known markers of undifferentiated cells, such as NANOG, OCT3/4, and SOX2, exhibited the same tendencies in the presence or absence of a feeder layer (Figure 1B). Next, we analyzed the GSL-glycan profiles of two undifferentiated iPSC lines (201B7 and 606A1) under feeder-free culture conditions. MALDI-TOF MS spectra of GSL-glycans on feeder-free iPSCs are shown in Figure 1C. The GSL-glycan profiles of both iPSC lines were very similar and stem cell-specific: high expression of globo- and (n) lacto-series GSLs, and low expression of ganglioside GSLs such as GM3 (Hex2NeuAc1: GSL-4), GM2 (Hex2HexNAc1NeuAc1: GSL-7), and GD3 (Hex2NeuAc2: GSL-11). Lacto-N-fucopentaose I (LNFP I; Hex3HexNAc1Fuc1: GSL-8) and fucosylated glycan (Hex4HexNAc1Fuc1: GSL-13), for which signals were observed in both iPSC lines, are as same as those on feeder iPSCs as previously described [20]. Based on these results, we used feeder-free iPSCs for further investigations of differentiation into progenitor cells (iPS-MSC like cells) and chondrocytes (iPS-MSC-CDs).

### 3.2. Evaluation of Pluripotent Stem Cell Markers and Gsl-Glycan Profiles of Chondrogenic Progenitor Cells (iPS-MSC Like Cells)

Next, we performed differentiation into chondrogenic progenitor cells (iPS-MSC like cells) as previously described [17] with minor modifications, using feeder-free iPSCs. Although cuboidal spindle-shaped cells were observed in iPS-MSC like cells through passage three, mesenchymal stem cell-like cells exhibited a heterogenous morphology, resembling a mixed population of elongated and cuboidal spindle-shaped cells. By passage six, iPS-MSC like cells showed a homogenous fibroblast-like morphology, matching the typical morphology of BMSCs (Figure 2A).

During differentiation of human iPSCs into iPS-MSCs, the expression levels of pluripotency markers (NANOG, OCT3/4, SOX2) were significantly reduced at passage three, approaching the levels in BMSCs (Figure 2B). By passage six, pluripotent stem cell markers were hardly expressed.

Cellular properties of iPS-MSC like cells (P6) and BMSCs (P3) were evaluated by flow cytometry using antibodies against CD surface antigens [23]. As shown in Figure 2C, the typical MSC-positive markers such as CD44 (>99.5%), CD73 (>99.9%), and CD105 (>98.0%) were highly expressed in all cells; however, CD90 was expressed at much lower levels on iPSC-MSC like cells than on BMSCs. Expression of typical MSC-negative markers, such as CD34, CD11b, CD19, CD45, and HLA-DR, was largely absent in iPSC-MSCs (all negative markers < 0.5%).

Next, we performed GSL-glycan analysis of iPS-MSC like cells and BMSCs using the same procedure described above. The levels of globo- and (n) lacto-series GSL-glycans diminished rapidly when iPSCs were induced to differentiate into iPS-MSC like cells, whereas expression of ganglioside GSLs such as GSL-4, GSL-7, and GSL-11 increased, consistent with previously reported results [14]. iPSC-specific fucosylated glycans (GSL-8 and GSL-13) were barely expressed (Figure 3). The detected GSL-glycans were almost identical between iPS-MSCs and BMSCs.

Together, these results suggest that (1) iPS-MSC like cells lost the properties of undifferentiated iPSCs, including expression of pluripotent stem cell markers and iPS-specific GSL-glycans, and (2) iPS-MSC like cells acquired properties that resembled the immunophenotype of BMSCs and consequently differentiated into mesenchymal stem cell-like cells.

### 3.3. Evaluation of Gsl-Glycan Profiles on iPS-MSC-CDs and Human Cartilage

Although transforming-growth-factor-β3 (TGF-β3) and BMP-2 are commonly used for chondrogenic differentiation [24], induction of BMP-2 is also reported for differentiation of mesenchymal stem cells into chondrocytes under micromass culture conditions [17]. In this study, we performed high-density micromass cultures of human BMSCs (P3) and iPS-MSC like cells (P6) using the induction growth factor of BMP-2 to produce hemispherical transparent cartilage particles. Sulfated proteoglycans were evaluated by Alcian blue histochemical staining on days 1, 7, 14, and 21. Alcian blue staining indicated gradual accumulation of sulfated proteoglycans and elevated compaction of the cells within the central core of the micromass in both BMSC-CDs and iPS-MSC-CDs (Figure 4).

Ultimately, by day 21 of micromass cultivation, hemispherical transparent cartilage particles approximately 1 to 3 mm in diameter were formed (Figure 4B).

Gene expression of chondrogenic markers such as *SOX9* (the master transcription factor of chondrogenesis), *COL2A1* (marker of matrix formation of hyaline cartilage that encodes type II collagen, a key chondrogenic marker), and *ACAN* (marker of matrix formation encoding for aggrecan) significantly increased over time during chondrogenic differentiation from both lines of iPS-MSC like cells. Meanwhile, expression of *COL1A1* (marker of matrix formation of fibrous cartilage encoding for type I collagen), *RUNX2* (osteogenic marker), and *PPARγ* (adipogenic marker) did not significantly vary at any time point (Figure 4C). In contrast, during chondrogenic differentiation of BMSCs, *COL1A1*, *PPARγ* and *RUNX2* increased under the same micromass culture conditions (Figure 4C).

Next, we performed GSL-glycan analysis of chondrogenic cells differentiated from iPS-MSC like cells on days 1, 7, 14, and 21. Human cartilage harvested from adult femoral head was used as a healthy control. Twenty-three signals corresponding to GSL-glycans were detected from chondrogenic cells and human cartilage, as summarized in Table 1. Before induction of chondrogenic differentiation, gangliosides such as GSL-4, GSL-7, GSL-12 (GM1: Hex_3_HexNAc_1_NeuAc_1_), and GSL-19 (GD1a: Hex_3_HexNAc_1_NeuAc_2_) were observed in iPS-MSCs of both cell lines (Figure 3). After day 1 of chondrogenic differentiation, levels of GSL-4 and GSL-7 decreased, whereas levels of GSL-11 and GSL-19 increased (Figure 4D). After 21 days of high-density micromass culture, the relative levels of GSL-4 and GSL-1 (Hex2) significantly decreased, whereas the levels of four GSL-glycans (GSL-2: Hex3, GSL-5: Hex3HexNAc1, GSL-12, and GSL-19) significantly increased (Table 1 and Appendix A). The relative levels of GSL-glycans in cartilage particles differentiated from the two iPSC lines that were highly correlated (r2 > 0.95) at all time points (Appendix A). During chondrogenic differentiation of BMSCs, GSL-4 drastically decreased at the same time as induction of iPS-MSC like cells into chondrocytes, whereas GSL-5 and GSL-11 showed different tendencies as shown in Appendix A. In human cartilage, the major GSL-glycans were GSL-2, GSL-4, GSL-5, GSL-11, GSL-12, and GSL-19 (Table 1 and Figure 4D). The GSL-glycan profile of human cartilage was closely similar to that of iPS-MSC-CDs at day 21 (Figure 5). Moreover, the correlation coefficients of GSL-glycans between human cartilage and iPS-MSC-CDs on day 21 were much higher than that between human cartilage and BMSCs (Figure 5).

## 4. Discussion

The purpose of this study was to clarify the characteristic alterations of cellular GSL-glycans during chondrogenic differentiation of human iPSCs. Our first objective was to ensure that human iPSCs accurately differentiate into chondrocytes. Although multiple reports have described induction of iPSCs into chondrocytes [25,26,27,28], we specifically investigated differentiation of iPSCs first into chondrogenic progenitor cells, followed by differentiation of the resultant cells into chondrocytes, under previously described micromass culture conditions [17]. In this study, we performed the chondrogenic differentiation from both human BMSCs and iPS-MSC like cells. iPS-MSC like cells showed a homogenous fibroblast-like morphology at passage six and were used for the cell evaluation. After initial induction of chondrogenic progenitor cells, expression levels of pluripotent stem cell markers (*NANOG*, *OCT3/4*, *SOX2*) were significantly reduced, and MSC-positive markers such as CD44, CD73, and CD105 were highly expressed on the surface of both BMSCs and iPS-MSC like cells. However, CD90 antigen, which plays an important role in maintaining the stemness of MSCs, was rarely expressed in iPS-MSC like cells in sharp contrast to that of BMSCs. It has been reported that the number of passages of human MSCs has effects on morphological, phenotypic, genetic changes, and different BMSC subpopulations in vitro [29,30]. The passage number of MSCs is very important and must be considered in the study of chondrogenic differentiation. Most of these results were consistent with the results reported by Guzzo except for CD90 [17]. These differences of CD90 expression may be attributed to a MEF feeder layer. Previously, Moraes et al. reported that MSCs shift from the undifferentiated state towards a state that is more susceptible to differentiation when the level of CD90 decreases [31]. These results suggest that iPS-MSCs without CD90 antigen may be in a highly susceptible differentiated state. After chondrogenic differentiation of iPS-MSC like cells under micromass culture conditions, the morphology of spindle-shaped cells was transformed to hyaline-like cartilage characterized by enrichment of chondrocyte specific *SOX9*, *COL2A1* and *ACAN*, as well as a lack of fibrous cartilage formation (*COL1A1*), osteogenesis (*RUNX2*), and adipogenesis (*PPARγ*). On the other hand, *COL1A1*, *PPARγ* and *RUNX2* increased with chondrogenic differentiation of BMSCs. Yang et al. reported that osteogenesis in human MSCs (P4) and MSCs (P8) was clearly different [30]. Although we did not measure the cellular markers depending on different passages, these results suggest that iPS-MSC like cells were superior in differentiation into chondrocytes than BMSCs. Consequently, we observed the generation of hyaline-like cartilage via the induction of mesenchymal-like progenitor cells from undifferentiated feeder-free human iPSCs.

Our second objective was to analyze the expression of cellular GSL-glycans during chondrogenic differentiation. GSL-glycans on the cell surface play important roles in biological events, and they vary widely corresponding to cell types and cell differentiations. Undifferentiated human iPSCs expressed high levels of globo- and (n) lacto-series GSLs, which shifted to ganglio-series GSLs, such as GM3, GM2, GM1, and GD1a, when iPSCs differentiated into iPS-MSCs. After chondrogenic differentiation of both BMSCs and iPS-MSC like cells, the expression of GM3 increased temporarily. Ryu et al. reported that GM3 and GD3 were expressed after chondrogenic differentiation and GM3 enhanced TGF-β signaling via SMAD 2/3 [32]. These results might suggest that BMSCs and iPS-MSCs differentiated into chondrocytes. However, GM3 levels markedly decreased, and GM1, GD3, and GD1 levels increased during further differentiation into chondrocytes. To evaluate the cell differentiation using GSL-glycome analysis, it is necessary to understand more about the relationship between GSL-glycans and various signaling pathways. Finally, the profile of chondrogenic cellular GSL-glycans came to resemble that of human cartilage. David et al. reported that the major gangliosides in osteoarthritis (OA) and control cartilage are GM3, GD3, and GD1a [33]. The results of our GSL-glycan analysis of human cartilage were in close agreement with previous reports, but also revealed that GM1 and GM1α containing α 2,6-linked sialic acid were present as gangliosides.

Epidermal growth factor receptor (EGFR) is a key representative of tyrosine kinase receptors, which are ubiquitous actors in cell signaling, proliferation, differentiation, and migration. GM3 in plasma membrane maintains EGFR in an inactive state and inhibits EGFR signaling [34]. Conversely, GD1a and GM1 promote EGFR activation [35,36]. Alteration of the ganglioside biosynthetic pathway may be a critical event in chondrogenic differentiation. Our results suggest that cellular GSL-glycan profiles could be used to evaluate different cell types and monitor the differentiation process. Although GSL-glycans differed between human cartilage and iPS-MSC-CDs, the overall GSL-glycan profile of human cartilage was very similar to that of iPS-MSC-CDs at day 21. Understanding the expression and structure of GSL-glycans based on the expression of a single gene would be difficult because GSL-glycans are elongated by a wide range of gene products that affect their biosynthesis. Therefore, identifying GSL-glycans could help to explain the mechanism of chondrogenic differentiation of iPSCs.

In this study, we evaluated chondrogenic differentiation from feeder-free iPSCs by detecting alterations in cellular GSL-glycans, in conjunction with genetic and histological evaluation. Elucidation of the mechanism responsible for these changes could provide insight into the differentiation mechanism of iPSCs, which is difficult to explain in terms of gene and protein levels.

## 5. Conclusions

In this study, we performed GSL-glycan analyses of iPSCs and their derivative cells. During differentiation into chondrocytes via iPSC-derived mesenchymal stem cell like cells, the profile of cellular GSL-glycans changed in a characteristic manner in each derivative cell type. Furthermore, alteration of GSL-glycans could be observed depending on the culture time of iPS-MSC-CDs. In particular, the levels of GM3 (GSL-4) drastically decreased during chondrogenic differentiation. On the other hand, Hex_3_HexNAc_1_ (GSL-5) of iPS-MSC-CDs significantly increased in sharp contrast to that of BMSC. GSL-11 gradually increased during differentiation into chondrocytes. The resultant GSL-glycan profile of iPS-MSC-CDs was very close to that of hyaline cartilage.

We demonstrated that the GSL-glycome profile could be used along with cell markers and cell-surface antigens to evaluate iPSCs and their derivative cells. Cellular GSL-glycomic analysis may facilitate the evaluation of target cells, leading to safer iPSC-based regenerative medicine.

## Figures and Tables

**Figure 1 biomolecules-10-01622-f001:**
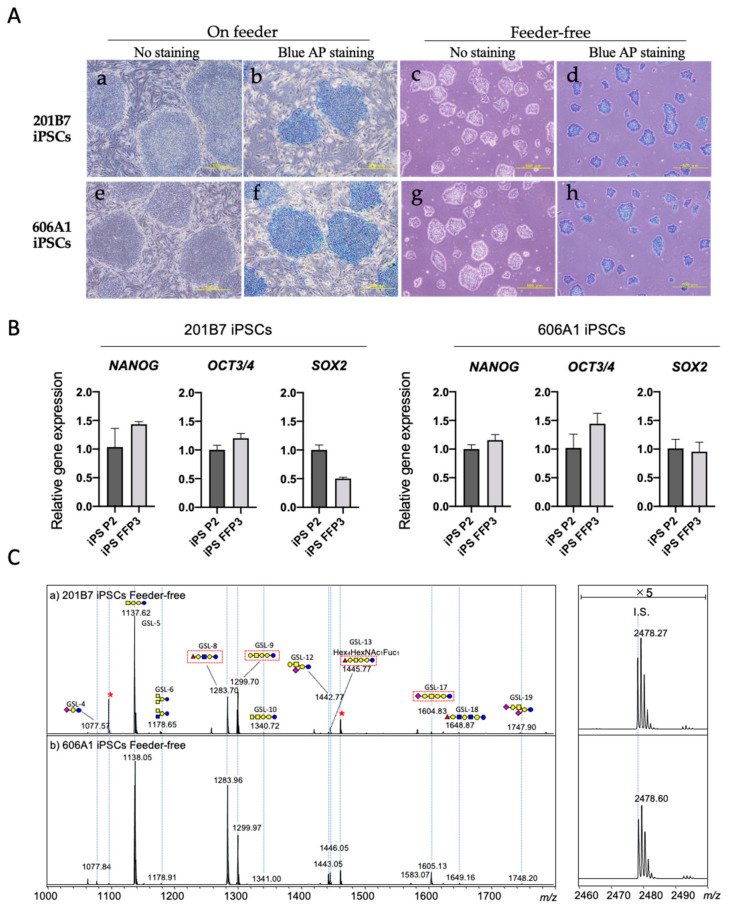
Evaluation of the pluripotent stem cell markers and GSL-glycan profiles of on feeder and feeder-free human iPSCs. (**A**) Representative microscopic findings of undifferentiated iPSCs 201B7 and 606A1. Both iPS cell lines formed typical colonies on feeder cells (a,e) and were stained with blue alkaline phosphatase (blue AP) (b,f). Under feeder-free conditions, typical iPSC colonies were observed (c,g) and confirmed by blue AP staining (d,h). Scale bars = 500 μm. (**B**) Gene expression analyses of pluripotent stem cell markers such as *NANOG*, *OCT3/4*, and *SOX2* in iPSCs on feeder at passage 2 (iPS P2), and feeder-free iPSCs at passage 3 (iPS FFP3). Values are presented as means ± SD. (**C**) MALDI-TOF MS spectra showing GSL-glycans on feeder-free human iPSCs 201B7 (a) and 606A1 (b). Red asterisks (*) and dashed red boxes indicate free oligosaccharides and undifferentiated iPSC-specific GSL-glycans, respectively.

**Figure 2 biomolecules-10-01622-f002:**
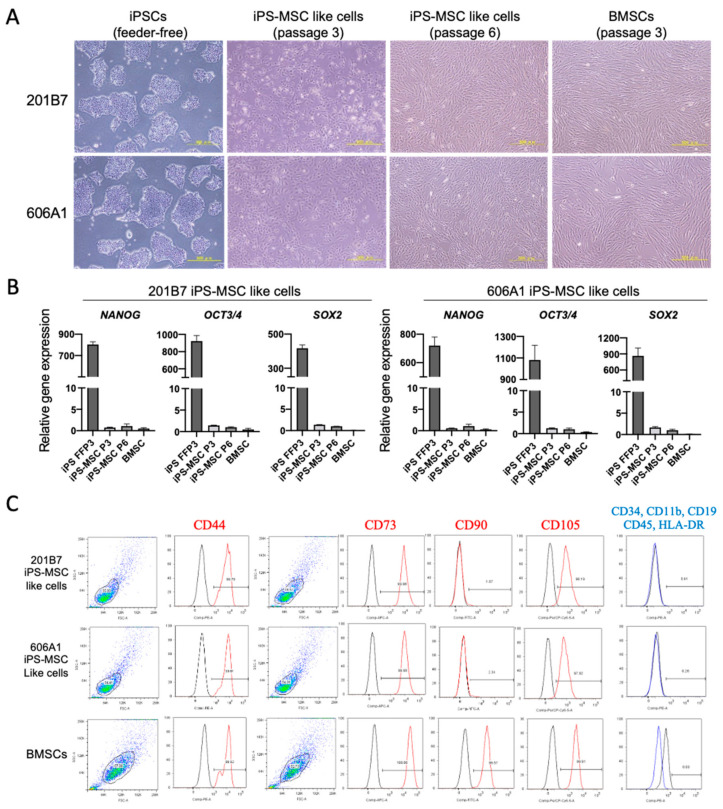
Disappearance of pluripotent stem cell markers and appearance of mesodermal markers during differentiation of iPSCs into iPSC-derived mesenchymal stem cell like cells (iPS-MSC). (**A**) Morphology of the iPS-MSC like cells differentiated from iPS cell lines (201B7 and 606A1) at passage 3 and 6, and human BMSCs at passage 3. Scale bars = 500 μm. (**B**) Gene expression analyses of pluripotent stem cell markers such as *NANOG*, *OCT3/4*, and *SOX2* in feeder-free iPSCs at passage 3 (iPS FFP3), iPS-MSCs at passages 3 and 6 (iPS-MSC P3 and P6), and BMSCs (passage 3). Values are presented as means ± SD. (**C**) Expression of surface antigens in iPS-MSC like cells derived from both iPSC lines, and BMSCs as determined by fluorescence activated cell sorter (FACS) analysis. Representative FACS profiles of iPS-MSC like cells (P6) and BMSCs (P3) associated with the mesenchymal phenotype (red font: positive antibodies against CD44, CD73, CD90, and CD105; blue font: cocktail of negative antibodies against CD34, CD11b, CD19, CD45, and HLA-DR). Red profiles: population stained with positive antibodies; blue profile: population stained with negative antibodies; black profile: population stained with isotype control.

**Figure 3 biomolecules-10-01622-f003:**
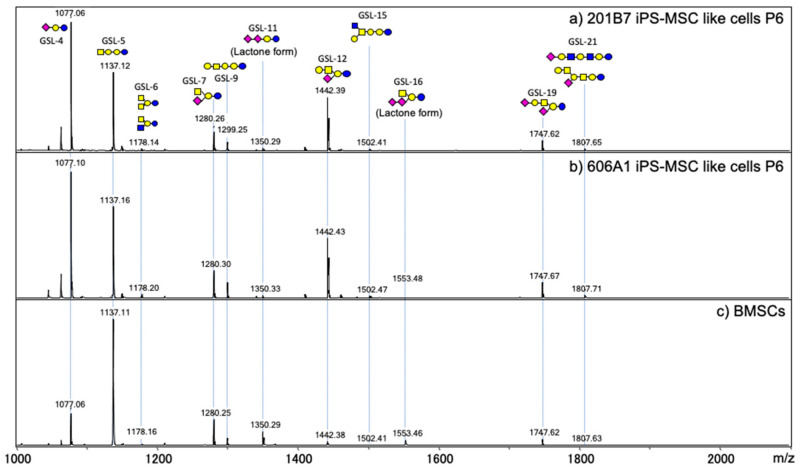
MALDI-TOF MS spectra of GSL-glycans. 201B7 iPS-MSC like cells (**a**), 606A1 iPS-MSC like cells (**b**), and human BMSCs (**c**).

**Figure 4 biomolecules-10-01622-f004:**
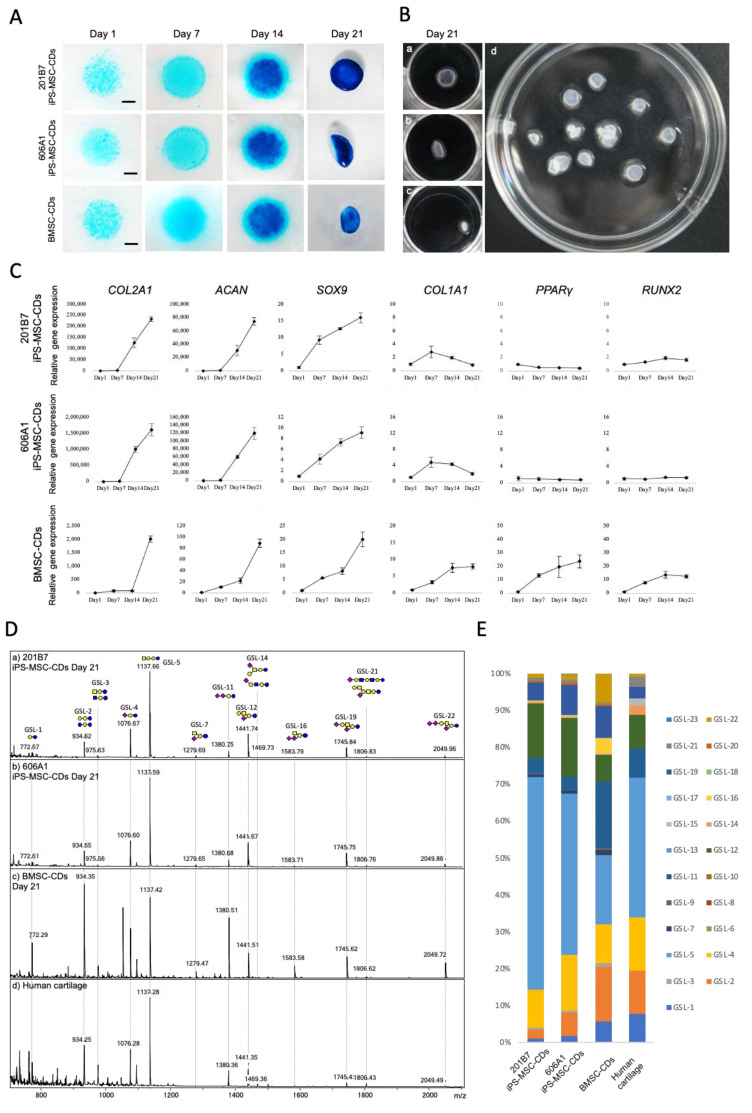
Evaluation of GSL-glycan profiles on iPS-MSC-CDs, BMSC-CDs, and human cartilage. (**A**) Alcian blue staining revealed gradual accumulation of proteoglycan-rich matrix during micromass cultures of iPS-MSCs and BMSCs. Scale bars = 1 mm. (**B**) Representative macroscopic findings of cartilage particles on day 21 of micromass cultures in a 24-well plate (a,b,c). a: 201B7, b: 606A1, c: collection of cartilage particles (606A1) in a 6 cm dish, d: hemispherical transparent cartilage particles. (**C**) qRT-PCR gene expression analyses of iPS-MSC-CDs (201B7 and 606A1) and BMSC-CDs on days 1, 7, 14, and 21 of chondrogenic differentiation. Expressions of chondrogenic markers (*SOX9*, *COL2A1*, *ACAN*, *and COL1A1*), adipogenic markers (*PPARγ*), and osteogenic markers (*RUNX2*) were evaluated. Values are presented as means ± SD. (**D**) MALDI-TOF MS spectra of GSL-glycans of 201B7 iPS-MSC-CDs at day 21 (a), 606A1 iPS-MSC-CDs at day 21 (b), BMSC-CDs at day 21 (c) and human cartilage (d). (**E**). Relative levels of GSL-glycans in 201B7 iPS-MSC-CDs at day 21, 606A1 iPS-MSC-CDs at day 21, BMSC-CDs at day 21 and human cartilage.

**Figure 5 biomolecules-10-01622-f005:**
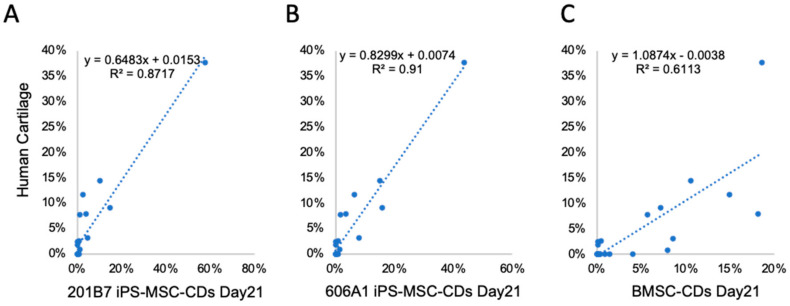
Correlation analysis between GSL-glycans of human cartilage and chondrogenic differentiated cells. (**A**) 201B7 iPS-MSC-CDs at day 21. (**B**) 606A1 iPS-MSC-CDs at day 21. (**C**) BMSC-CDs at day 21.

**Table 1 biomolecules-10-01622-t001:** List of GSL-glycans in 201B7 iPS-MSC-CDs, 606A1 iPS-MSC-CDs, BMSC-CDs, and human cartilage.

No.	Glycan Composition	m/z	201B7 iPS-MSC-CDs				606A1 iPS-MSC-CDs				BMSC-CDs				Normal Cartilage
			Day 1	Day 7	Day 14	Day 21	Day 1	Day 7	Day 14	Day 21	Day 1	Day 7	Day 14	Day 21	
GSL-1	(Hex)2	772.34	3.29%	2.97%	1.41%	1.05%	3.08%	2.70%	1.77%	1.82%	5.37%	4.51%	7.04%	5.66%	7.76%
GSL-2	(Hex)3	934.39	0.68%	2.41%	3.47%	2.51%	0.28%	2.98%	5.94%	6.30%	10.73%	14.44%	15.43%	14.94%	11.70%
GSL-3	(Hex)2(HexNAc)1	975.42	1.75%	0.48%	0.23%	0.40%	2.35%	0.48%	0.67%	0.49%	1.47%	0.42%	1.22%	0.88%	0.00%
GSL-4	(Hex)2(α2,3NeuAc)1	1076.46	71.38%	24.48%	8.91%	10.31%	60.13%	29.45%	11.90%	15.17%	32.20%	23.92%	10.92%	10.58%	14.52%
GSL-5	(Hex)3(HexNAc)1	1137.47	8.35%	45.44%	53.41%	57.66%	11.09%	39.89%	46.42%	43.68%	12.42%	20.44%	25.05%	18.58%	37.79%
GSL-6	(Hex)2(HexNAc)2	1178.50	0.26%	0.00%	0.00%	0.00%	0.46%	0.00%	0.00%	0.00%	0.26%	0.15%	0.29%	0.22%	0.00%
GSL-7	(Hex)2(HexNAc)1(α2,3NeuAc)1	1279.54	0.78%	0.00%	0.85%	0.79%	1.02%	0.19%	1.09%	0.86%	8.20%	3.43%	1.68%	1.39%	0.00%
GSL-8	(Hex)3(HexNAc)1(Fuc)1	1283.53	0.00%	0.00%	0.00%	0.00%	0.00%	0.00%	0.00%	0.00%	0.00%	0.00%	0.00%	0.00%	0.00%
GSL-9	(Hex)4(HexNAc)1	1299.52	0.41%	0.35%	0.32%	0.32%	1.48%	0.25%	0.17%	0.19%	1.70%	0.53%	0.41%	0.35%	0.00%
GSL-10	(Hex)3(HexNAc)2	1340.55	0.00%	0.00%	0.00%	0.00%	0.06%	0.07%	0.03%	0.06%	0.12%	0.05%	0.04%	0.11%	0.00%
GSL-11	(Hex)2(α2,3NeuAc)2	1380.59	2.33%	5.86%	4.00%	4.03%	3.92%	6.85%	4.35%	3.48%	19.23%	10.77%	11.67%	18.14%	7.95%
GSL-12	(Hex)3(HexNAc)1(α2,3NeuAc)1	1441.60	7.93%	12.58%	14.55%	14.96%	11.54%	12.16%	14.92%	15.95%	0.84%	5.35%	8.29%	7.22%	9.20%
GSL-13	(Hex)4(HexNAc)1(Fuc)1	1445.58	0.00%	0.00%	0.00%	0.00%	0.00%	0.00%	0.00%	0.00%	0.00%	0.00%	0.00%	0.00%	0.00%
GSL-14	(Hex)3(HexNAc)1(α2,6NeuAc)1	1469.63	0.13%	0.30%	0.21%	0.20%	0.23%	0.24%	0.18%	0.18%	0.07%	0.20%	0.19%	0.15%	2.48%
GSL-15	(Hex)4(HexNAc)2	1502.60	0.38%	0.34%	0.10%	0.03%	0.52%	0.25%	0.15%	0.10%	0.09%	0.09%	0.22%	0.13%	1.87%
GSL-16	(Hex)2(HexNAc)1(α2,3NeuAc)2	1583.67	0.00%	0.18%	0.77%	0.30%	0.05%	0.11%	0.63%	0.38%	5.08%	3.49%	5.39%	4.07%	0.00%
GSL-17	(Hex)4(HexNAc)1(α2,3NeuAc)1	1603.60	0.00%	0.11%	0.17%	0.19%	0.00%	0.12%	0.16%	0.17%	0.25%	0.29%	0.35%	0.29%	0.00%
GSL-18	(Hex)4(HexNAc)2(Fuc)1	1648.66	0.00%	0.00%	0.00%	0.00%	0.00%	0.00%	0.00%	0.00%	0.00%	0.00%	0.00%	0.00%	0.00%
GSL-19	(Hex)3(HexNAc)1(α2,3NeuAc)2	1745.72	0.37%	2.33%	8.17%	4.82%	0.64%	2.31%	8.52%	8.07%	1.35%	8.05%	6.63%	8.57%	3.18%
GSL-20	(Hex)3(HexNAc)1(α2,3NeuAc)(α2,6NeuAc)	1773.75	0.78%	0.56%	0.57%	0.46%	1.45%	0.50%	0.51%	0.51%	0.12%	0.52%	0.41%	0.32%	0.00%
GSL-21	(Hex)4(HexNAc)2(α2,3NeuAc)1	1806.73	1.18%	1.15%	0.96%	0.71%	1.64%	1.13%	0.90%	0.98%	0.09%	0.23%	0.70%	0.48%	2.66%
GSL-22	(Hex)3(HexNAc)1(α2,3NeuAc)3	2049.90	0.00%	0.37%	1.81%	1.26%	0.00%	0.26%	1.67%	1.52%	0.42%	3.14%	4.07%	7.96%	0.90%
GSL-23	(Hex)5(HexNAc)3(α2,3NeuAc)1	2171.80	0.00%	0.11%	0.07%	0.00%	0.06%	0.06%	0.03%	0.11%	0.00%	0.00%	0.00%	0.00%	0.00%

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
