# Peer review of "Alterations of Glycosphingolipid Glycans and Chondrogenic Markers during Differentiation of Human Induced Pluripotent Stem Cells into Chondrocytes"

_biomolecules, 2020, doi:10.3390/biom10121622_

Round 1

Reviewer 1 Report

This is a good and reliable study of glycolipid glycome in the process of stem cell differentiation. The presented data in themselves are not so significant for this area, but their value becomes high due to the finding of difference in glycomes in the process of cell differentiation. The "dynamic" approach increases the reliability of the results (since the original cells serve as the best control) and provides food for thought about the functional significance of the disappearance/decrease of some glycans and the appearance/elevation of others. I would like to see in the Discussion section the authors' thoughts on this score (functional significance), albeit crude and speculative, this would increase the reader's interest in this publication and in the topic in general. This is my only wish for the article, there is no significant criticism.

Author Response

We greatly appreciate the Reviewer’s insightful comments, which have helped us to improve the quality of our manuscript. Our point-by-point responses to each comment are shown below.

This is a good and reliable study of glycolipid glycome in the process of stem cell differentiation. The presented data in themselves are not so significant for this area, but their value becomes high due to the finding of difference in glycomes in the process of cell differentiation. The "dynamic" approach increases the reliability of the results (since the original cells serve as the best control) and provides food for thought about the functional significance of the disappearance/decrease of some glycans and the appearance/elevation of others. I would like to see in the Discussion section the authors' thoughts on this score (functional significance), albeit crude and speculative, this would increase the reader's interest in this publication and in the topic in general. This is my only wish for the article, there is no significant criticism.

Response: We thank the reviewer for valuable comment. Although we don’t have answer about the functional significance of some glycan alteration, their value is important to evaluate the process of cell differentiation. So, we revised the follow sentences about the relationship between glycans and signaling in the discussion section.

Page 11, Line 345-Page 12, Line 353: “After chondrogenic differentiation of both BMSCs and iPS-MSC like cells, the expression of GM3 increased temporarily. Ryu et al. reported that GM3 and GD3 were expressed after the chondrogenic differentiation and GM3 enhanced TGF-β signaling via SMAD 2/3 [30]. These results might suggest that BMSCs and iPS-MSCs differentiated into chondrocytes. However, GM3 levels markedly decreased, and GM1, GD3, and GD1 levels increased during further differentiation into chondrocytes. To evaluate the cell differentiation using GSL-glycome analysis, it is a necessary to understand more about the relationship between GSL-glycans and various signaling pathway. Finally, the profile of chondrogenic cellular GSL-glycans came to resemble that of human cartilage.”

Reviewer 2 Report

The manuscript entitled "Alterations of glycosphingolipid glycans and chondrogenic markers during differentiation of human induced pluripotent stem cells into chondrocytes" describes the ability of mesenchymal stem cells (MSCs) that are derived from induced pluripotent stem cells (iPSCs) to differentiate to become chondrocytes. The authors have performed experiments that reveal certain histological and genetic markers that are specifically related to several glycans that are increased or decreased during differentiation of those iPSCs-MSCs into chondrocytes, and their results were comparable to that of the native source of MSCs.

There are some issues in the manuscript

  1. Please mention clearly the difference between the chondrocytes that are differentiated from native MSCs as well as from iPSCs, in what do they have their own advantages and disadvantages? How are they clinically relevant?
  2. Page 2 line 48 mentions about chondrocyte proliferation, chondrocytes are already terminally differentiated, can they proliferate again? Please clarify.
  3. MSCs and iPSC MSCs were used at different passages for flow cytometry assay, since there are differences in marker expression between the passages, looking at the differences in CD90 for example, the subpopulations of MSCs (BMSCs vs iPSC-MSC) are different. How this would influence on the chondrocyte characterization and function? Please add discussion about the different subpopulations of MSCs based on their marker characterization.
  4. Page 4 - Please decipher ALP
  5. Page 6 - Line 215 mentions that the MSCs were used at P6, however, the methods section mentions as P3.

Overall, this study described the differentiation of MSCs from iPSCs which is not novel. To make it more interesting and publishable, please add a discussion of the consequences of the alterations observed during chondrocytic differentiation and what would be the advantages to use such methods from iPSCs and not just from BMSCs.

Author Response

We greatly appreciate the Reviewer’s insightful comments, which have helped us to improve the quality of our manuscript. Our point-by-point responses to each comment are shown below.

The manuscript entitled "Alterations of glycosphingolipid glycans and chondrogenic markers during differentiation of human induced pluripotent stem cells into chondrocytes" describes the ability of mesenchymal stem cells (MSCs) that are derived from induced pluripotent stem cells (iPSCs) to differentiate to become chondrocytes. The authors have performed experiments that reveal certain histological and genetic markers that are specifically related to several glycans that are increased or decreased during differentiation of those iPSCs-MSCs into chondrocytes, and their results were comparable to that of the native source of MSCs.

There are some issues in the manuscript:

Comment 1: Please mention clearly the difference between the chondrocytes that are differentiated from native MSCs as well as from iPSCs, in what do they have their own advantages and disadvantages? How are they clinically relevant?

Response: We thank your valuable comment. To explain the difference between BMSC-CDs and iPS-MSC-CDs, we added the experimental data of 1) Alcian blue staining, 2) qRT‐PCR gene expression analyses, and 3) MALDI-TOF MS spectra of BMSC-CDs in revised Figure 4. Basing on our PCR results, the property of iPS-MSC-CDs is closer to that of articular cartilage than that of BMSC-CDs. Moreover, the GSL-glycan profile of iPS-MSC-CDs at day 21 was closely similar to that of human cartilage rather than BMSC-CDs. These results indicated that the ability of chondrogenic differentiation of iPS-MSCs might be superior to that of BMSC in vitro. We added the follow sentences in our manuscript.

Page 8, Line 258-266: Although transforming-growth-factor-β (TGF-β3) and BMP-2 are commonly used for chondrogenic differentiation [24], induction of BMP-2 are also reported for differentiation of mesenchymal stem cells into chondrocytes under micromass culture conditions [17]. In this study, we performed high-density micromass cultures of human BMSCs (P3) and iPS-MSC like cells (P6) using the induction growth factor of BMP-2 to produce hemispherical transparent cartilage particles. Sulfated proteoglycans were evaluated by Alcian blue histochemical staining on days 1, 7, 14, and 21. Alcian blue staining indicated gradual accumulation of sulfated proteoglycans and elevated compaction of the cells within the central core of the micromass in both BMSC-CDs and iPS-MSC-CDs (Fig. 4A).

Page 10, Line 288-289: In contrast, during chondrogenic differentiation of BMSCs, COL1A1, PPARγ and RUNX2 increased under the same micromass culture conditions (Fig. 4C).

Page 10, Line 306- Page 11, Line 312: During chondrogenic differentiation of BMSCs, GSL-4 drastically decreased as same as induction of iPS-MSC like cells into chondrocytes, whereas GSL-5 and GSL-11 showed different tendency as shown in Fig. S3. In human cartilage, the major GSL-glycans were GSL-2, GSL-4, GSL-5, GSL-11, GSL-12, and GSL-19 (Table 1 and Fig. 4D). The GSL-glycan profile of human cartilage was closely similar to that of iPS-MSC-CDs at day 21 (Fig.5). Moreover, the correlation coefficients of GSL-glycans between human cartilage and iPS-MSC-CDs on day 21 were much higher than that between human cartilage and BMSCs (Fig. 5).

Page 11, Line 332-340: After chondrogenic differentiation of iPS-MSC like cells under micromass culture conditions, the morphology of spindle-shaped cells was transformed to hyaline-like cartilage characterized by enrichment of SOX9, COL2A1 and ACAN, as well as lack of COL1A1, osteogenesis (RUNX2), and adipogenesis (PPARγ). On the other hand, COL1A1 and RUNX2 increased when chondrogenic differentiation of BMSCs. These results suggested that iPS-MSC like cells were superior in differentiation into chondrocytes than BMSCs. Consequently, we observed the generation of hyaline-like cartilage via the induction of mesenchymal-like progenitor cells from undifferentiated feeder-free human iPSCs.

Comment 2: Page 2 line 48 mentions about chondrocyte proliferation, chondrocytes are already terminally differentiated, can they proliferate again? Please clarify.

Response: As you said, cartilage is already terminally differentiated tissue and have a low self-repair ability. However, chondrocyte has also been indicated to be cultured and proliferated in vitro. In clinical applications, autologous chondrocyte implantation (ACI) has a 30-year history and is an established technique for the treatment of cartilage defects. It involves an initial cartilage biopsy, from which chondrocytes are cultured in vitro. We revised the follow sentence in our manuscript.

Page 2, Line 46-49: but several obstacles to wider adoption persist: the need to sacrifice healthy cartilage, the requirement for a two-step surgery with an initial harvest of cartilage, the difficulty of acquiring sufficient numbers of cells, and the low proliferation capacity of chondrocytes in cartilage implantation [3].

Comment 3: MSCs and iPSC MSCs were used at different passages for flow cytometry assay, since there are differences in marker expression between the passages, looking at the differences in CD90 for example, the subpopulations of MSCs (BMSCs vs iPSC-MSC) are different. How this would influence on the chondrocyte characterization and function? Please add discussion about the different subpopulations of MSCs based on their marker characterization.

Response: Thank you very much for this important comment. Please allow us to explain for the different subpopulations of MSCs in this study. Basing on the results reported by Guzzo et al, the cellular morphology and property of iPS-MSCs (passage 5-7) were most close to that of MSCs, and iPS-MSCs (P6) were used for cell differentiation and flow cytometry assay [17]. In this study, iPS-MSC like cells were induced from the original iPSCs and iPS-MSC like cells at passage 6 showed similar cellular morphology of BMSCs. So, iPS-MSC like cells (P6) were used for the cell evaluation. To avoid de-differentiation, BMSCs at passage 3 were used as control. Cellular properties of iPS‐MSC like cells (P6) and BMSCs (P3) were evaluated by flow cytometry using antibodies against CD surface antigens and most of CDs were consistent with the results reported by Guzzo except for CD90. In order to discuss their CD makers in detail, we revised the follow sentences in our manuscript.

Page 11, Line 323-331: After initial induction of chondrogenic progenitor cells, expression levels of pluripotent stem cell markers (NANOG, OCT3/4, SOX2) were significantly reduced, and MSC-positive markers such as CD44, CD73, and CD105 were highly expressed on the surface of both BMSCs and iPS-MSC like cells. However, CD90 antigen, which play an important role in maintaining the stemness of MSCs, was rarely expressed in iPS-MSC like cells in sharply contrast with that of BMSCs. Most of these results were consistent with the results reported by Guzzo except for CD90 [17]. These differences of CD90 expression may be attributed to a murine embryonic fibroblast (MEF) feeder layer. Previously, Moraes et al. reported that MSCs shift from the undifferentiated state toward a state that is more susceptible to differentiation when the level of CD90 decreases [29].

Comment 4: Page 4 - Please decipher ALP

Response: Thank you for this comment. We revised the follow sentence in our manuscript. We also replaced the word “ALP” with “blue AP” through our manuscript.

Page 3, Line 100-102: Under these conditions, iPSCs remained in an undifferentiated state, as assessed by blue alkaline phosphatase (Blue AP) staining (Vector® Blue AP Substrate kit, SK-5300; Vector Labs, Burlingame, CA, USA).

Comment 5: Page 6 - Line 215 mentions that the MSCs were used at P6, however, the methods section mentions as P3.

Response: Thank you for this comment. We are sorry for your confusing. We used the iPS-MSC like cells at both P3 and P6 to evaluate the morphology. We revised the follow sentences.

Page 3, Line 116-118: iPS-MSC like cells at passage 6 were used as the final iPS-derived mesenchymal stem cell-like cells. MSCs (PT-2501, Lonza) derived from bone marrow were pre-cultured and passaged three times and used as control.

Page 6, Line 221-222: By passage 6, iPS-MSC like cells showed a homogenous fibroblast-like morphology, matching the typical morphology of BMSCs (Fig. 2A).

Comment 6: Overall, this study described the differentiation of MSCs from iPSCs which is not novel. To make it more interesting and publishable, please add a discussion of the consequences of the alterations observed during chondrocytic differentiation and what would be the advantages to use such methods from iPSCs and not just from BMSCs.

Response: Thank you for this comment. We added the experimental data of 1) Alcian blue staining, 2) qRT‐PCR gene expression analyses, and 3) MALDI-TOF MS spectra of BMSC-CDs in revised Figure 4. The results in this study indicated the property of cartilage-like tissues derived from iPS-MSCs is very similar to that of articular cartilage with the significantly elevated of hyaline-cartilage formation markers (SOX9, COL2A1, and ACAN) and the suppressed of fibrous-cartilage formation marker (COL1A1), osteogenic marker (RUNX2), and adipogenic marker (PPARγ) over time during chondrogenic differentiation. However, the same specific changes of those markers were not observed during the chondrogenic differentiation of BMSCs. Moreover, the correlation coefficients of GSL-glycans between human cartilage and iPS-MSC-CDs on day 21 were much higher than that between human cartilage and BMSCs These results suggested that MSC-like cells derived from iPS cells maybe have a higher potential for chondrogenic differentiation than BMSCs in vitro.Thus, our responses are similar to comment 1.

Reviewer 3 Report

Why did the authors choose to use feeder-free model? Should it contribute to increasing efficiency of differentiation in micromass culture?

Why did the authors take advantage of BMP-2 instead of  e.g. TGF-β3 which is known as the growth factor with the most chondrogenic potential?

In my judgement, the authors should perform a second independent method for confirmation of obtaining MSC-like cells (apart from flow cytometry analysis). 

Perhaps, authors could add some short sentences and appriopriate references involving gene expression profile of hiPSCs undergoing chondrogenic differentiation. This could enrich introduction/discussion.

Description of Figure 4: I do not see in the figure presentation of accumulation of proteoglycan-rich matrix in BMSCs (A). I see only iPS-MSC-CDs. Similarly, I do not see gene expression of material isolated from BMSC-CDs.

Description of Figure 5: I suggest clarify that correlation analysis concerns GSL-glycan profile. 

Lines 308-309: Could the authors elaborate on this idea?

Table 1 and/or conlusions. I suggest to more emphasize the application nature of obtained results: e.g. underline in the table the most important GSL-glycans: GSL-2, GSL-4, GSL-5, GSL-11, GSL-12 and GSL-19 or add in the conclusion that they can be used in the future as biomarkers (or more adequate word).

Author Response

We greatly appreciate the Reviewer’s insightful comments, which have helped us to improve the quality of our manuscript. Our point-by-point responses to each comment are shown below.

Comment 1: Why did the authors choose to use feeder-free model? Should it contribute to increasing efficiency of differentiation in micromass culture?

Response: Thank you for this comment. We considered that iPSCs cultured on feeder layer might increase the risk of MEF contamination. Comparing with iPS-MSC-CDs from on feeder iPSCs reported previously, chondrocyte specific markers were high expression and GSL-glycan profile was very close to that of hyaline cartilage in feeder-free model. We revised the follow sentence in our manuscript.

Page 4, Line 185-187: Although maintenance of human induced pluripotent stem cells (iPSCs) typically requires the absence of a murine embryonic fibroblast (MEF) feeder layer to further enhance human iPSC adhesion and growth, iPSCs cultured on feeder layer might increase the risk of MEF contamination.

Comment 2: Why did the authors take advantage of BMP-2 instead of e.g. TGF-β3 which is known as the growth factor with the most chondrogenic potential?

Response: Thank you for this valuable comment. Although TGF-β3 is commonly used for chondrogenic differentiation recently, there are also many reports about induction of BMP-2 [26]. . It is well established that BMP signaling exerts multiple stage-specific effects on cartilage development [Kramer et al., 2000; Pizette and Niswander, 2000; Yoon and Lyons, 2004]. Guzzo et al. demonstrated that hyaline cartilage can be obtained by using BMPs and we also chose the protocol of BMP-2. The investigation of the TGF-β3 induced chondrogenic differentiation is for further study. We revised the follow sentence in our manuscript.

Page 8, Line 258-266: Although transforming-growth-factor-β (TGF-β3) and BMP-2 are commonly used for chondrogenic differentiation [24], induction of BMP-2 are also reported for differentiation of mesenchymal stem cells into chondrocytes under micromass culture conditions [17]. In this study, we performed high-density micromass cultures of human BMSCs (P3) and iPS-MSC like cells (P6) using the induction growth factor of BMP-2 to produce hemispherical transparent cartilage particles. Sulfated proteoglycans were evaluated by Alcian blue histochemical staining on days 1, 7, 14, and 21. Alcian blue staining indicated gradual accumulation of sulfated proteoglycans and elevated compaction of the cells within the central core of the micromass in both BMSC-CDs and iPS-MSC-CDs (Fig. 4A).

Comment 3: In my judgement, the authors should perform a second independent method for confirmation of obtaining MSC-like cells (apart from flow cytometry analysis).

Response: Thank you very much for this constructive comment. We think it is helpful for us to explain the properties of MSC-like cells. However, we are really sorry that it’s very difficult for us to confirm the MSC-like cells with a second independent method within this short revising period. To explain the difference between BMSC and iPS-MSC like cells, we added the experimental data of flow cytometry using BMSCs (P3) in revised Figure 2 and revised the follow sentences in results and discussion part.

Page 10, Line 288-289: In contrast, during chondrogenic differentiation of BMSCs, COL1A1, PPARγ and RUNX2 increased under the same micromass culture conditions (Fig. 4C).

Page 11, Line 323-331: After initial induction of chondrogenic progenitor cells, expression levels of pluripotent stem cell markers (NANOG, OCT3/4, SOX2) were significantly reduced, and MSC-positive markers such as CD44, CD73, and CD105 were highly expressed on the surface of both BMSCs and iPS-MSC likes. However, CD90 antigen, which play an important role in maintaining the stemness of MSCs, was rarely expressed in iPS-MSC like cells in sharply contrast with that of BMSCs. Most of these results were consistent with the results reported by Guzzo except for CD90 [17]. These differences of CD90 expression may be attributed to a murine embryonic fibroblast (MEF) feeder layer. Previously, Moraes et al. reported that MSCs shift from the undifferentiated state toward a state that is more susceptible to differentiation when the level of CD90 decreases [29].

Comment 4: Perhaps, authors could add some short sentences and appropriate references involving gene expression profile of hiPSCs undergoing chondrogenic differentiation. This could enrich introduction/discussion.

Response: Thank you for this comment. According to reviewer’s comment, we revised and added the follow sentences in introduction part.

Page 2, Line 57-65: Recency, gong et al indicated that NFATc1 (nuclear factor of activated T-cells) may interact with the master transcriptional regulator Sox9 and be involved in chondrogenesis regulation [8]. However, Sfougataki et al reported that type Ⅱ collagen was major production in BMSC derived micromasses, in contrast, it was not detect in PSC-MSC derived micromasses [9]. Thus, evaluation of chondrogenic differentiation by genetic markers alone may not be sufficient. Despite the histological analysis of extracellular matrix (ECM) components are also performed usually for evaluation of cartilage formation [7], it remains unclear what extent of matching is required for acceptable allogeneic transplantation of chondrocytes or cartilage.

Comment 5: Description of Figure 4: I do not see in the figure presentation of accumulation of proteoglycan-rich matrix in BMSCs (A). I see only iPS-MSC-CDs. Similarly, I do not see gene expression of material isolated from BMSC-CDs.

Response: Thank you for this detailed comment. We added the experimental data of 1) Alcian blue staining, 2) qRT‐PCR gene expression analyses, and 3) MALDI-TOF MS spectra of BMSC-CDs in revised Figure 4. Basing on our PCR results, the property of iPS-MSC-CDs is closer to that of articular cartilage than that of BMSC-CDs. Moreover, the GSL-glycan profile of iPS-MSC-CDs at day 21 was similar to that of human cartilage rather than BMSC-SDs. These results indicated that the efficiency of chondrogenic differentiation of iPS-MSCs might be superior to that of BMSC in vitro. We added the follow sentences in our manuscript.

Page 8, Line 258-266: Although transforming-growth-factor-β (TGF-β3) and BMP-2 are commonly used for chondrogenic differentiation [24], induction of BMP-2 are also reported for differentiation of mesenchymal stem cells into chondrocytes under micromass culture conditions [17]. In this study, we performed high-density micromass cultures of human BMSCs (P3) and iPS-MSC like cells (P6) using the induction growth factor of BMP-2 to produce hemispherical transparent cartilage particles. Sulfated proteoglycans were evaluated by Alcian blue histochemical staining on days 1, 7, 14, and 21. Alcian blue staining indicated gradual accumulation of sulfated proteoglycans and elevated compaction of the cells within the central core of the micromass in both BMSC-CDs and iPS-MSC-CDs (Fig. 4A).

Page 10, Line 288-289: In contrast, during chondrogenic differentiation of BMSCs, COL1A1, PPARγ and RUNX2 increased under the same micromass culture conditions (Fig. 4C).

Page 10, Line 306-Page 11, Line 312: During chondrogenic differentiation of BMSCs, GSL-4 drastically decreased as same as induction of iPS-MSC like cells into chondrocytes, whereas GSL-5 and GSL-11 showed different tendency as shown in Fig. S3. In human cartilage, the major GSL-glycans were GSL-2, GSL-4, GSL-5, GSL-11, GSL-12, and GSL-19 (Table 1 and Fig. 4D). The GSL-glycan profile of human cartilage was closely similar to that of iPS-MSC-CDs at day 21 (Fig.5). Moreover, the correlation coefficients of GSL-glycans between human cartilage and iPS-MSC-CDs on day 21 were much higher than that between human cartilage and BMSCs (Fig. 5).

Page 11, Line 332-340: After chondrogenic differentiation of iPS-MSC like cells under micromass culture conditions, the morphology of spindle-shaped cells was transformed to hyaline-like cartilage characterized by enrichment of SOX9, COL2A1 and ACAN, as well as lack of COL1A1, osteogenesis (RUNX2), and adipogenesis (PPARγ). On the other hand, COL1A1, PPARγ and RUNX2 increased when chondrogenic differentiation of BMSCs. These results suggested that iPS-MSC like cells were superior in differentiation into chondrocytes than BMSCs. Consequently, we observed the generation of hyaline-like cartilage via the induction of mesenchymal-like progenitor cells from undifferentiated feeder-free human iPSCs.

Comment 6: Description of Figure 5: I suggest clarify that correlation analysis concerns GSL-glycan profile.

Response: Thank you for this comment. We revised the legend in Figure 5.

Figure 5. Correlation analysis between GSL-glycans of human cartilage and chondrogenic differentiated cells. A. 201B7 iPS-MSC-CDs at day 21. B. 606A1 iPS-MSC-CDs at day 21. C. BMSC-CDs at day 21.

Comment 7: Lines 308-309: Could the authors elaborate on this idea?

Response: We are sorry that our discussion here wasn’t clear enough. MSC-positive markers such as CD44, CD73, and CD105 were highly expressed both BMSCs and iPS-MSCs. However, CD90 antigen was rarely expressed in iPS-MSCs in sharply contrast with that of BMSCs. Most of these results were consistent with the results reported by Guzzo except for CD90. Thus, we replaced the word “iPS-MSCs” with “iPS-MSC like cells” through our manuscript.

These differences of CD90 expression may be attributed to feeder-free iPSCs without a murine embryonic fibroblast (MEF) feeder layer. We added the follow sentence in our discussion section.

Page 11, Line 328-329: These differences of CD90 expression may be attributed to a murine embryonic fibroblast (MEF) feeder layer.

Comment 8: Table 1 and/or conclusions. I suggest to more emphasize the application nature of obtained results: e.g. underline in the table the most important GSL-glycans: GSL-2, GSL-4, GSL-5, GSL-11, GSL-12 and GSL-19 or add in the conclusion that they can be used in the future as biomarkers (or more adequate word).

Response: Thank you for this important suggestion. In order to emphasize the application nature of obtained results, we added the Fig. S3 GSL-glycan alterations (GSL-4, -5, and -11) during chondrogenic differentiation and revised the follow sentence in our conclusion.

Page 12, Line 378-382: In particular, the levels of GM3 (GSL-4) drastically decreased during chondrogenic differentiation. On the other hand, Hex3HexNAc1 (GSL-5) of iPS-MSC-CDs significantly increased in sharply contrast with that of BMSC. GSL-11 gradually increased during differentiation into chondrocytes. The resultant GSL-glycan profile of iPS-MSC-CDs was very close to that of hyaline cartilage.

Round 2

Reviewer 2 Report

Thank you for your answers and edits on the original manuscript.

Though the authors have mentioned and cited a couple of papers that have shown similar results with MSCs markers at different passage numbers, the therapeutic efficacy of the MSCs at different passages may be different. Therefore, it is encouraged to always have the exact same passage number to compare the markers expression. Having almost same sub population of MSCs at different passage numbers may not be an ideal cell population that will have the same therapeutic effects.  Please add a section to discuss this factor which would also help answering another previous comment to elaborate on the following question "Please add discussion about the different subpopulations of MSCs based on their marker characterization".

With those changes, this manuscript can be published.

Author Response

Response: We appreciate your valuable comment. We discussed about the different subpopulations of MSCs and revised the follow sentences in our discussion section.

Page 11, Line 322-347: In this study, we performed the chondrogenic differentiation from both human BMSCs and iPS-MSC like cells. iPS-MSC like cells showed a homogenous fibroblast-like morphology at passage 6 and used for the cell evaluation. After initial induction of chondrogenic progenitor cells, expression levels of pluripotent stem cell markers (NANOG, OCT3/4, SOX2) were significantly reduced, and MSC-positive markers such as CD44, CD73, and CD105 were highly expressed on the surface of both BMSCs and iPS-MSC likes. However, CD90 antigen, which play an important role in maintaining the stemness of MSCs, was rarely expressed in iPS-MSC like cells in sharply contrast with that of BMSCs. It has been reported that the number of passages of human MSCs have effects on morphological, phenotypic, genetic changes, and different BMSC subpopulations in vitro [29,30]. The passage number of MSCs is a very important and must be considered in the study of chondrogeninc differentiation. Most of these results were consistent with the results reported by Guzzo except for CD90 [17]. These differences of CD90 expression may be attributed to a murine embryonic fibroblast (MEF) feeder layer. Previously, Moraes et al. reported that MSCs shift from the undifferentiated state toward a state that is more susceptible to differentiation when the level of CD90 decreases [31]. These results suggest that iPS-MSCs without CD90 antigen may be in a highly susceptible differentiated state. After chondrogenic differentiation of iPS-MSC like cells under micromass culture conditions, the morphology of spindle-shaped cells was transformed to hyaline-like cartilage characterized by enrichment of chondrocyte specific SOX9, COL2A1 and ACAN, as well as lack of fibrous cartilage formation (COL1A1, osteogenesis (RUNX2), and adipogenesis (PPARγ). On the other hand, COL1A1, PPARγ and RUNX2 increased when chondrogenic differentiation of BMSCs. Yang et al. reported that osteogenesis in human MSCs (P4) and MSCs (P8) was clearly different [30]. Although we didn’t measure the cellular markers depending on different passages, these results suggested that iPS-MSC like cells were superior in differentiation into chondrocytes than BMSCs. Consequently, we observed the generation of hyaline-like cartilage via the induction of mesenchymal-like progenitor cells from undifferentiated feeder-free human iPSCs.

Reviewer 3 Report

Thank you for improving your manuscript and addressing all the comments.

Author Response

We greatly appreciated the your time reviewing and helping us to improve the manuscript.